# Extreme Serum Titanium Concentration Induced by Acetabular Cup Failure: Unveiling a Unique Scenario of Titanium Alloy Debris Accumulation

**DOI:** 10.3390/bioengineering11030235

**Published:** 2024-02-28

**Authors:** Samo K. Fokter, Živa Ledinek, Milka Kljaić Dujić, Igor Novak

**Affiliations:** 1Clinical Department of Orthopedic Surgery, University Medical Centre Maribor, Ljubljanska 5, 2000 Maribor, Slovenia; igor.novak@ukc-mb.si; 2Faculty of Medicine, University of Maribor, Slomškov trg 15, 2000 Maribor, Slovenia; 3Faculty of Medicine, University of Ljubljana, Kongresni trg 12, 1000 Ljubljana, Slovenia; 4Department of Pathology, University Medical Centre Maribor, Ljubljanska 5, 2000 Maribor, Slovenia; ziva.ledinek@ukc-mb.si; 5Department of Radiology, University Medical Centre Maribor, Ljubljanska 5, 2000 Maribor, Slovenia; milka.kljaicdujic@ukc-mb.si

**Keywords:** total hip arthroplasty, titanium alloy, orthopedic implants, implant failure, adverse local tissue reactions

## Abstract

The majority of contemporary total hip arthroplasty (THA) implants are constructed from Ti alloys, which are generally believed to generate fewer adverse local tissue reactions (ALTRs) compared to CoCr alloys. This study presents a case of unusual primary THA failure where a substantial release of Ti alloy debris was observed. A 52-year-old active male underwent THA after post-traumatic aseptic necrosis of the femoral head in 2006. Seventeen years after the procedure, the patient presented with groin pain and a restricted range of motion. X-rays revealed the protrusion of the alumina ceramic head through the Ti_6_Al_4_V acetabular cup. Trace element analysis indicated significantly elevated levels of serum Ti, Al, and V. CT and MRI confirmed Ti alloy cup failure and a severe ALTR. During revision surgery, it was found that the worn-out ceramic head was in direct contact with the acetabular cup, having protruded through a central hole it had created over time. No acetabular liner was found. Histological analysis of his tissue samples showed wear-induced synovitis with areas of multinucleated foreign body giant cells and the accumulation of numerous metal particles but no acute inflammatory response. Six months after the revision THA, the patient has experienced favourable outcomes. This case provides an instructive illustration for studying the consequences of the substantial release of Ti alloy debris from orthopedic implants.

## 1. Introduction

With an aging global population, the prevalence of total hip replacements has surged, exceeding 330,000 procedures annually in the United States alone [1]. Contemporary total hip arthroplasty (THA) implants predominantly utilize titanium alloys for the structural components due to their favourable properties, including excellent biocompatibility, high strength-to-weight ratio, and superior corrosion resistance, ensuring their suitability for long-term implantation into the human body [2]. While titanium alloys are preferred for their reduced risk of adverse local tissue reactions (ALTRs) compared to cobalt–chromium (CoCr) alloys, issues such as trunnionosis have been identified, linking titanium to pain, ALTRs, aseptic loosening, and fractures of the femoral prosthesis neck [3,4,5].

The articulation between the femoral and acetabular components in THA can be made from various material combinations, including metal-on-polyethylene (MoP), ceramic-on-polyethylene (CoP), metal-on-metal (MoM), and ceramic-on-ceramic (CoC). Each pairing offers distinct advantages and considerations, with the choices often guided by patient-specific factors such as age, activity level, and potential for allergic reactions. 

The longevity of THA is inversely related to patient age at the time of the initial surgery. Data from the New Zealand Joint Registry (NZJR) indicate a lifetime revision risk of 27.6% for patients aged 46 to 50 years, which significantly decreases to 1.1% for those aged 90 to 95 years [6]. Aseptic loosening emerges as the predominant reason for THA revision, with the youngest patients showing the lowest 10-year implant survival rate [6]. Furthermore, systematic reviews, including one by Prokopetz et al., highlight male gender as a risk factor for revision due to aseptic loosening and infection. Factors such as younger age, diagnosis of avascular necrosis, and larger femoral head size have also been identified as predictors of increased revision risk [7].

Despite case reports indicating pseudo-tumors and ALTRs without CoCr implants, the research into the effects of titanium alloy debris has been largely concentrated on primary THA cases with bimodular stems. These stems are characterized by an exchangeable neck connected to the femoral stem via an additional Morse taper, leading to increased dissemination of metal degradation products into the surrounding tissues compared to standard single-modular stem systems [8,9,10,11].

However, even in bimodular stem THA systems, the release of titanium alloy debris is typically gradual, resulting in low blood concentrations of trace elements in patients with well-functioning implants [12,13]. Conversely, suboptimal THA performance may lead to increased titanium alloy debris release, provoking a heightened tissue and immune response [14]. The objective of this study is to meticulously document and analyze a singular case of primary THA malfunction, distinguished by an extensive and previously unrecorded long-term release of titanium alloy debris. This investigation not only fills a gap in the existing medical literature by reporting on the scale of titanium release in such malfunctions but also aims to elucidate the potential clinical implications and mechanisms underlying this phenomenon. The novelty of this work lies in its detailed examination of an exceptionally high concentration of titanium detected post-THA failure and its subsequent management, providing invaluable insights into the challenges and considerations of diagnosing and treating implant-related complications.

## 2. Case Presentation

A 35-year-old male, actively engaged in farming, began experiencing progressive hip pain in 2006 following a bull kick. Subsequently, he underwent osteosynthesis for a femoral neck fracture using a Dynamic Hip Screw (DHS, Synthes GmbH, Oberdorf, Switzerland). Aseptic necrosis of the femoral head ensued, leading to the removal of the osteosynthetic material eight months later. During the same surgical session, the patient underwent THA, where a 52 mm porous Ti6Al4V alloy press-fit acetabular cup (Trilogy AB, Zimmer, Warsaw, IN, USA) with an aluminum oxide ceramic shell insert (BIOLOX Forte, CeramTec, Plochlingen, Germany—distributed by Zimmer, Warsaw, IN, USA) was combined with a #18 Versys femoral stem (Zimmer, Warsaw, IN, USA). The femoral stem featured a Ti6Al4V alloy collarless fiber metal taper with a standard neck offset. It was coupled with a 28 mm alumina ceramic femoral head (BIOLOX Forte, CeramTec—distributed by Zimmer, Warsaw, IN, USA). This coupling was achieved using a 12/14 mm Morse taper, with the femoral head featuring a neck length designated as “0 mm” to indicate a neutral, or medium, position. This neutral neck length does not alter the leg length or femoral offset, serving as a standard configuration for achieving balanced anatomical dimensions post-surgery. The patient’s postoperative course was uneventful, with discharge instructions to use crutches for six weeks, followed by unrestricted weight bearing.

Despite the absence of pain, the patient reported unusual sounds in the hip during weight bearing, linked to the CoC bearing. Standard radiographs of the right hip were deemed normal, as depicted in the first available postoperative X-ray (Figure 1). Therefore, no further evaluation or specific treatment was advised for the patient. 

Seventeen years post-THA, the now 52-year-old patient presented to our outpatient clinics with groin pain and a severely limited range of motion. The patient’s Harris Hip Score (HHS) was 48 points. His laboratory parameters, including white cell count, hemoglobin level, liver and kidney function, erythrocyte sedimentation rate (ESR), and C-reactive protein (CRP) level, were within the normal limits (Table 1). Separate blood samples were taken and stored in special containers (BD Vacutainer^®^ Trace Element, Becton Dickinson, Franklin Lakes, NJ, USA). His serum levels of titanium, aluminium, and vanadium were determined at the nationally accredited laboratory (Clinical Institute for Clinical Chemistry and Biochemistry, Ljubljana, Slovenia) for trace element analysis. Measurements of titanium in his blood samples were performed using inductively coupled plasma mass spectroscopy (ICP-MS) with the Octopole Reaction System (7700×, Agilent Technologies, Santa Clara, CA, USA). X-ray and computer tomography (CT) evaluation revealed the protrusion of the alumina ceramic head through the Ti6Al4V acetabular cup (Figure 2), necessitating immediate scheduling for THA revision due to the severe condition of the patient’s hip and the significant risk of further complications.

During the revision surgery, the absence of an acetabular liner was noted, and the worn-out ceramic head had made direct contact with the acetabular cup, protruding through a central hole formed over time (Figure 3). Pseudo-tumors in the soft tissue and severe metallosis near the implant were excised, with tissue samples sent for histological analysis and cultures. Acetabular screws underwent sonication. The stable femoral stem with an intact Morse taper for femoral neck–head coupling necessitated only the exchange of the acetabular component and femoral head. A 58 mm Ti6Al4V press-fit acetabular cup (DELTA TT, LimaCorporate, Udine, Italy) with an aluminum oxide matrix composite ceramic liner (BIOLOX Delta, CeramTec) was inserted and secured with three screws. A 36 mm aluminum oxide matrix composite ceramic revision femoral head with a Ti6Al4V sleeve (BIOLOX Delta Option, CeramTec) was attached to the original femoral stem, followed by repositioning (Figure 4). 

The soft tissues obtained during revision surgery were formalin-fixed, underwent routine tissue processing, and were paraffin-embedded. Out of each paraffin-embedded tissue block, 3–5 µm thick paraffin sections were cut and placed on glass slide (Superfrost, Waldemar Knittel Glasbearbeitungs GmbH, Braunschweig, Germany). Hematoxylin and eosine (HE) staining was performed on an automated Ventana HE 600 system (Ventana Medical Systems, Roche, Basel, Switzerland). Microphotographs of the tissue slides were taken with an Aperio ScanScope CS (Aperio, Leica Biosystems, Nußloch, Germany) pathology slide scanner with 20× scanning magnification and exported using Aperio’s free image viewer (ImageScope (version: 12.4.6.5003), Leica Biosystems). 

Magnetic resonance imaging (MRI) of the patient’s hip was performed using a 1.5 T MRI machine (MAGNETOM Sola, Siemens, Munich, Germany). The protocol parameters were as follows: STIR TSE sequence; slice thickness: 3 mm; body coil; flip angle: 150°; TR: 3790 ms; TE: 35 ms. The wound healed uneventfully, and the patient was discharged on the ninth postoperative day. After a 10-day incubation period, all samples were found to be sterile. At the six-month follow-up after the revision THA, the patient reported favorable outcomes. The patient’s HHS was 97 points. His laboratory parameters, including white cell count, hemoglobin level, and liver and kidney function, as well as ESR and CRP level, were within the normal limits (Table 1). The radiograph of the patient’s hip depicted a stable THA construct without additional osteolysis (Figure 5).

### 2.1. Serum Trace Element Analysis

Serum trace element analysis was performed on the day before revision surgery, at 4-month follow-up (titanium only), and at 6-month follow-up. The trace element analysis indicated significantly elevated levels of serum titanium (norm < 6.0 μg/L), aluminium (norm < 10.0 μg/L), and vanadium (norm < 0.14 μg/L) at all times. The serum levels of titanium peaked to 1237.6 μg/L at 4 months after revision surgery. At the 6-month follow-up, the serum levels of titanium and vanadium were steady declining. However, the aluminium levels remain elevated and were even inclining at 6-month follow-up regarding the basic level before revision surgery (Table 2). The values for cobalt and chromium were in the normal range at all times (norm < 0.65 μg/L for cobalt and <0.75 μg/L for chromium). 

### 2.2. Histological Analysis

Tissue samples of the periprosthetic soft tissue that gave a macroscopic impression of the pseudo-tumors were sent to the pathology lab for histological analysis. The tissue samples were of irregular shapes, soft, greyish–brown to black in color, and up to 4 cm in size. Microscopic examination showed areas of necrotic tissue and fibrin depositions admixed with abundant metallic microparticles (Figure 6). There was extensive infiltration predominantly with macrophages, containing metallic microparticles and macroparticles, as well as areas of granulomatous inflammatory reaction with foreign body giant cells. Lymphoplasmacytic infiltrate was present but there were no granulocytes, indicating the absence of acute inflammation and therefore excluding infection.

After joint replacement surgery, regenerated synovial tissue and the periprosthetic membrane are referred to using the term “synovial-like interface membrane” (SLIM). The histological findings of SLIM pathology are classified into groups that aid the recognition of underlying pathophysiological mechanisms leading to implant failure. Based on the International Expanded SLIM Consensus Classification, the patient had wear-induced synovitis/a wear-induced SLIM, leading to an adverse local tissue reaction to the implant wear particles (type VI) with a predominantly macrophagic pattern with an absent or minimal lymphocytic response [15]. The patient did not meet criteria for periprosthetic joint infection (no neutrophils present in the tissue). 

### 2.3. MR Imaging

Prior to the revision surgery, MRI revealed substantial soft tissue alterations around the endo-prosthesis, particularly in the muscles proximal to the greater trochanter of the femur. Primarily, these changes manifested as solid chronic granulation with a smaller cystic component that was partially calcified. In both the described soft tissue changes and the periprosthetic osteolytic regions, minute hypointense inclusions of metal microparticles were observed (Figure 7a). Six months post-revision surgery, the previously extensive granulation changes in the surrounding soft tissues were diminished. There was reduced soft tissue edema, and the indications of local irritation had also decreased. The presence of slightly increased fluid around the endoprosthesis indicated residual seroma following the surgery (Figure 7b).

## 3. Discussion

The etiology behind the catastrophic failure of the titanium alloy acetabular cup remains elusive. Although the original manufacturer’s sticker, containing all pertinent data on the implant, was located in the patient’s medical records, no evidence of the acetabular liner was discovered during revision surgery. Additionally, despite employing various radiographic procedures (X-rays, CT, and MRI), we were unable to identify fragments or remnants of the ceramic liner. It is plausible that the entire acetabular liner was ground into tiny and fine-grain debris; however, such debris was not identified in the histological samples obtained during revision surgery. Furthermore, the absence of larger pieces of alumina ceramic liner, which would likely have eluded the grinding process between the intact alumina ceramic femoral head and the dome of the acetabular cup, speaks against this hypothesis. Conversely, it would be highly unusual for an orthopedic surgeon not to insert the acetabular liner into the acetabular cup. In such a scenario, over the years, the harder ceramic femoral head would inevitably create a void in the softer titanium alloy acetabular cup, particularly in a young and physically active patient. 

To the best of our knowledge, this investigation presents the highest documented serum concentration of titanium in a patient with a failed orthopedic implant. In patients with loosened titanium hip replacement components, elevated serum titanium levels are observed due to mechanical wear and corrosion at the interface between the implant and the surrounding bone. This phenomenon is primarily attributed to the increased friction resulting from implant loosening, leading to the generation of microscopic titanium particles. These particles, released into the surrounding tissues, may subsequently enter the bloodstream, causing an elevation in serum titanium levels. Furthermore, the body’s inflammatory response to these foreign particles exacerbates the situation by promoting further bone resorption and implant loosening. Corrosion processes, inherent to the physiological environment and exacerbated by the mechanical instability of the loosened implant, also contribute to the release of titanium ions into the serum. Thus, elevated serum titanium levels in such scenarios serve as a biomarker for the degree of implant wear and corrosion, reflecting the complex interplay between mechanical degradation and biological response. Previous studies exploring this phenomenon have typically been limited in scale, employed various implant designs, and utilized diverse analytical methods to assess the titanium content in the blood or serum of affected patients, contributing to the observed variability [16,17,18]. Instances of fractured ceramic liners in CoC prostheses have also been demonstrated to result in heightened systemic titanium levels. Swiatkowska et al. documented a case within their institution where such an occurrence was associated with a blood titanium level of 21.5 μg/L just before the revision [19]. The substantial increase was linked to the mechanical abrasion between the femoral head and fragments of the fractured ceramic liner, as well as the contact between the titanium femoral neck and the edge of the acetabular cup [19]. In addition to the observed increase in serum titanium levels, Sakamoto et al. have conclusively demonstrated the capacity of titanium particles to give rise to symptomatic solid pseudo-tumors [20].

Titanium is a key material in various medical devices, including joint implants, and has been associated with wear and corrosion, releasing inflammatory by-products into tissues and blood [21]. Unlike cobalt, titanium accumulates in the organs, posing a risk of tissue injury [22]. Despite emerging safety concerns, titanium’s impact remains understudied. The mechanisms, species (ions, particles), and cellular fate of the titanium released from implants are unclear. Establishing baseline and toxic titanium levels in biological fluids is challenging due to limited reliable detection methods. While high-resolution instruments show promise for blood titanium measurement, their accessibility remains restricted to specialized laboratories, hindering routine monitoring. Inconsistencies in titanium measurements can arise due to the absence of standardized protocols across laboratories, complicating inter-study comparisons. Recent findings by Koller and colleagues underscored significant variations in the results when identical baseline pooled blood samples were analyzed by different laboratories [23]. These discrepancies were attributed to differences in sample preparation, instrument types, and analytical approaches. Given these challenges, it is recommended to employ a single laboratory for titanium testing to ensure direct comparability, as values from different laboratories may not align accurately. In our investigation, we adhered to these guidelines and utilized a solitary laboratory accredited for titanium testing.

Based on the histological particle algorithm proposed by Perino et al., the microparticles found in tissue samples are mostly likely conventional non-ferrous metallic particles due to implant wear [24]. The described microparticles and macroparticles were greyish to intensively black, which is consistent with metal. Ceramic particles could also be present with a blackish color if oxidized and could also be abundant but usually present only as microparticles except in cases of prosthesis fracture, where larger microparticles have been described as well. Other implant wear particles and endogenous particles (such as hemosiderin) differ from metal particles in shape, size, and color or are birefringent. 

Several researchers have validated the occurrence of a deleterious pseudo-tumor post-THA even in the absence of an MoM bearing [25,26]. Notably, these instances involved the utilization of metal heads in conjunction with titanium alloy femoral stems. The responsibility of cobalt and chromium in ALTR generation has been well established [27,28]. Bisseling et al. reported a case involving a patient who underwent revision THA with a double-mobility MoP bearing two years after the primary procedure due to the presence of a soft tissue mass at the posterolateral aspect of the greater trochanter [29]. During the initial THA, an uncemented titanium–niobium (Ti-6Al-7Nb) stem (Zweymuller Alloclassic; Zimmer Orthopaedics, Warsaw, IN, USA) was implanted, featuring a 12/14 mm trunnion combined with an XXL (+10.5 mm) 28 mm cobalt–chromium head with a 12/14 mm tapered bore (Biomet, Warsaw, IN, USA). Despite chromium levels below the detection threshold and unreported serum titanium levels, elevated cobalt serum levels (5.7 µg/L) were observed. The authors concluded that a robust patient-specific immunological response to a modest quantity of metal debris and corrosion by-products resulting from the mismatched head–neck junction prompted the early revision. Emphasizing the potential role of the head–neck junction in generating metal debris and corrosion by-products, which may contribute to ALTRs, the authors cautioned against intermixing THA components from diverse manufacturers [29]. It is noteworthy that in our case, components exclusively from the same manufacturer were employed, and none contained cobalt or chromium.

Several researchers have advocated for the assessment of serum titanium levels in individuals with titanium-based orthopedic prostheses due to their potential as a diagnostic marker for implant performance [30,31]. This recommendation stems from the observation that implants, particularly those displaying looseness or signs of wear, including polyethylene wear-through, often release higher concentrations of titanium compared to well-functioning implants. Jacobs et al. demonstrated that elevated serum titanium levels could function as an indicator of patellar component failure or accelerated femoral component wear in total knee replacements with titanium alloy bearings [32]. Notably, patients with failed patellar components exhibited titanium concentrations approximately 50 times higher than those with intact components, with no discernible differences in aluminum and vanadium levels. Despite the authors acknowledging the unknown toxicological implications of these findings, we, too, did not observe any adverse effects of potential titanium toxicity in our patient.

Titanium, characterized by poor solubility, tends to accumulate in organ tissues such as the liver and spleen, while a fraction is excreted in the urine. Monitoring renal function may be crucial, but our patient showed no signs of impaired excretion processes. We attributed the skewed results to significant titanium release from the poorly functioning implant. A temporary rise in the serum titanium concentration at the 4-month follow-up was likely due to additional titanium release from the soft tissue pseudo-tumors, only partially accessible for excision. Despite titanium being considered less toxic than cobalt, our case highlights the potential for adverse tissue reactions in the absence of CoCr alloy components. Unfortunately, no titanium-chelating agents, comparable to those recently developed for cobalt, are currently available [33].

The unique aspect of our case lies in the extensive and early material wear observed, which is significantly influenced by the patient’s demographics—a young, active male. Such demographics typically correlate with higher activity levels and engagement in physically demanding tasks, factors known to increase mechanical demands on implants, thereby accelerating wear. Konopitski et al.’s meta-analysis and systematic review highlight how advances in surgical techniques and implant technology, such as the shift from metal on ultra-high molecular weight polyethylene (UHMWPE) to modern THA using ceramic on highly cross-linked polyethylene (HXLPE), have improved implant longevity [34]. Specifically, younger patients fitted with modern vitamin E-stabilized HXLPE liners show no increased revision rates at mid-term follow-up [35]. Additionally, Vendittoli et al.’s long-term prospective study underscores the benefits of CoC bearings in an active, younger patient demographic, where wear-related reactions pose significant concerns [36].

In the context of our patient, who engaged in demanding agricultural work, the higher mechanical loads expectedly expedited wear rates. This effect was exacerbated by the critical omission of a ceramic acetabular liner in the initial THA, a departure from best practices that significantly contributed to the accelerated wear of the titanium acetabular cup by the alumina ceramic head. Liners are essential for distributing load and minimizing friction between implant components; their absence in this case led to abnormal wear patterns and the observed failure.

This study encompasses several limitations that merit attention. A primary limitation is the absence of a comprehensive analysis of the severe metallosis observed in the tissues surrounding the implant. Given that the sole metal component implanted in our patient was the titanium alloy (Ti6Al4V), it is reasonable to infer that the metal debris identified consisted of titanium, aluminum, and vanadium. This inference is further supported by blood trace element analysis, which revealed elevated levels of titanium, aluminum, and vanadium. Despite this logical deduction, the direct analysis of tissue debris to confirm its composition was not conducted, representing a significant gap in our investigation. It is worth noting that the presence of such metal debris and its composition have been demonstrated in several previous studies [37,38,39,40].

Secondly, it is important to highlight that while elevated serum titanium levels were observed, indicating wear and fragmentation of the titanium alloy due to the interaction with alumina ceramic particles, this study did not directly measure titanium concentration in the surrounding and damaged tissues. Consequently, attributing the adverse tissue reactions solely to the accumulation of high concentrations of titanium in the tissues without specific tissue level analysis may oversimplify the complex etiology of these reactions. Future studies involving direct measurement of metal ion levels in the tissue surrounding failed implants could provide a more definitive link between local titanium accumulation and specific adverse tissue responses, further elucidating the biological mechanisms at play.

## 4. Conclusions

This case highlights the significant release of titanium alloy debris from dysfunctional orthopedic implants, revealing unprecedentedly high blood titanium concentrations as determined using ICP-MS. Notably, the blood titanium concentration surged after the acetabular cup revision, returning to approximately the level observed at the time of revision by the 6-month follow-up. This instance underscores the critical role of meticulous surgical techniques in arthroplasty and suggests the absence of a ceramic liner, despite its documentation in the surgery report. The improbability of the liner being reduced to fine-grain debris, as evidenced by the intact condition of the femoral head made of the same material, points toward a discrepancy in the initial surgical procedure. Despite these findings, no adverse health effects were observed in the patient. However, the significant titanium exposure, persisting at the 6-month follow-up, necessitates ongoing vigilance for any potential long-term health implications.

## Figures and Tables

**Figure 1 bioengineering-11-00235-f001:**
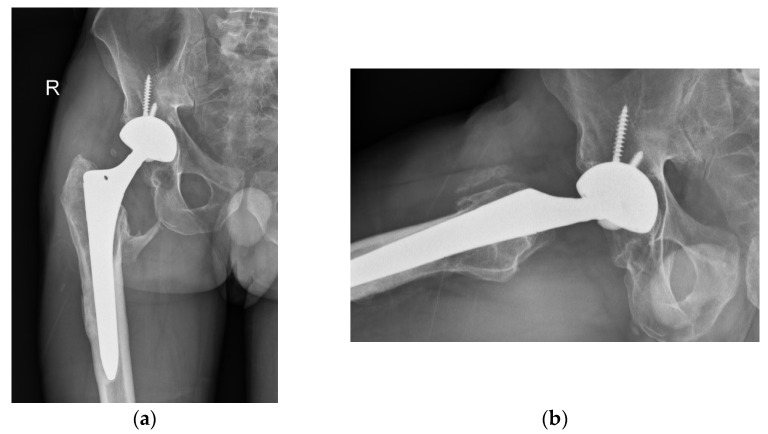
Radiographs of the patient’s right hip 8 years after primary THA. (**a**) Antero-posterior view; (**b**) lateral view. Note that the head is penetrating a lot into the cup (**a**) and that there is an offset of the ceramic head within the cup (**b**). Some heterotopic ossification at the level of the lesser trochanter is likely related to the patient’s previous osteosynthesis for a femoral neck fracture.

**Figure 2 bioengineering-11-00235-f002:**
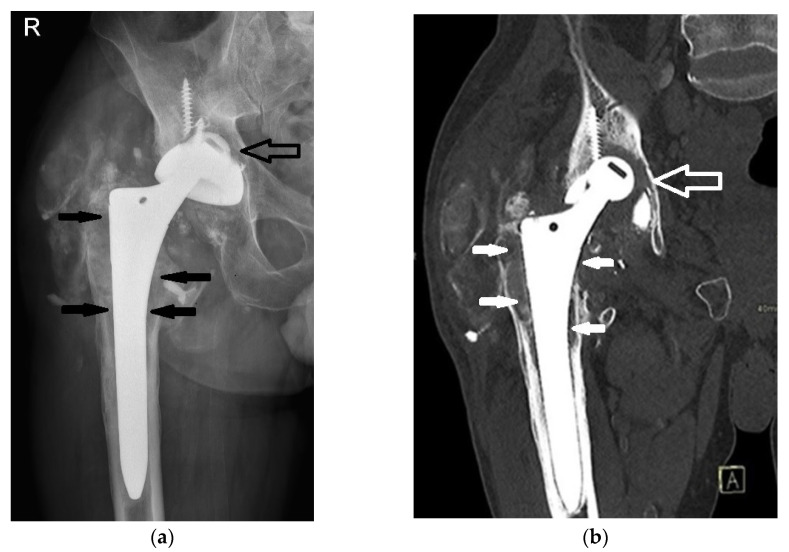
Radiographic imaging of the patient’s right hip 17 years after primary THA. Note alumina ceramic femoral head protrusion through the acetabular cup (hollow arrows), extensive calcifications in the soft tissues, and osteolysis (solid arrows). (**a**) Antero-posterior X-ray; (**b**) computer tomography (CT) scan with coronal reconstruction.

**Figure 3 bioengineering-11-00235-f003:**
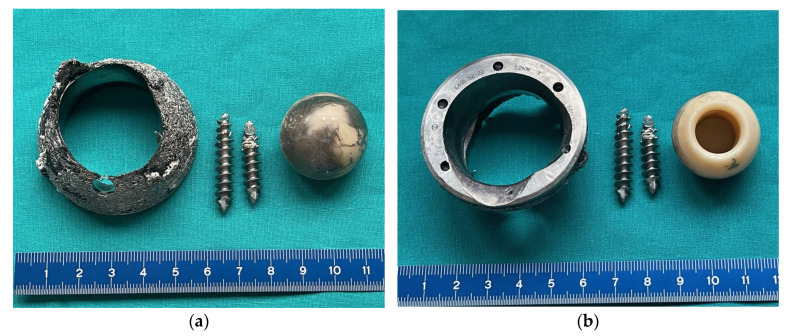
A photograph of the removed Ti6Al4V acetabular cup showing a central hole which was created by the alumina ceramic head. (**a**) Cranial view; (**b**) caudal view.

**Figure 4 bioengineering-11-00235-f004:**
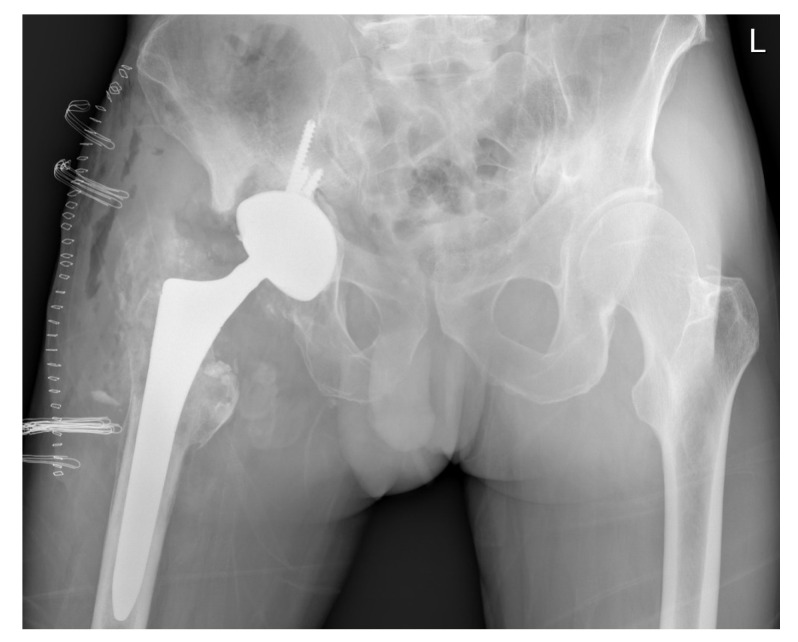
A radiograph of both hips depicting patient’s right hip immediately after revision.

**Figure 5 bioengineering-11-00235-f005:**
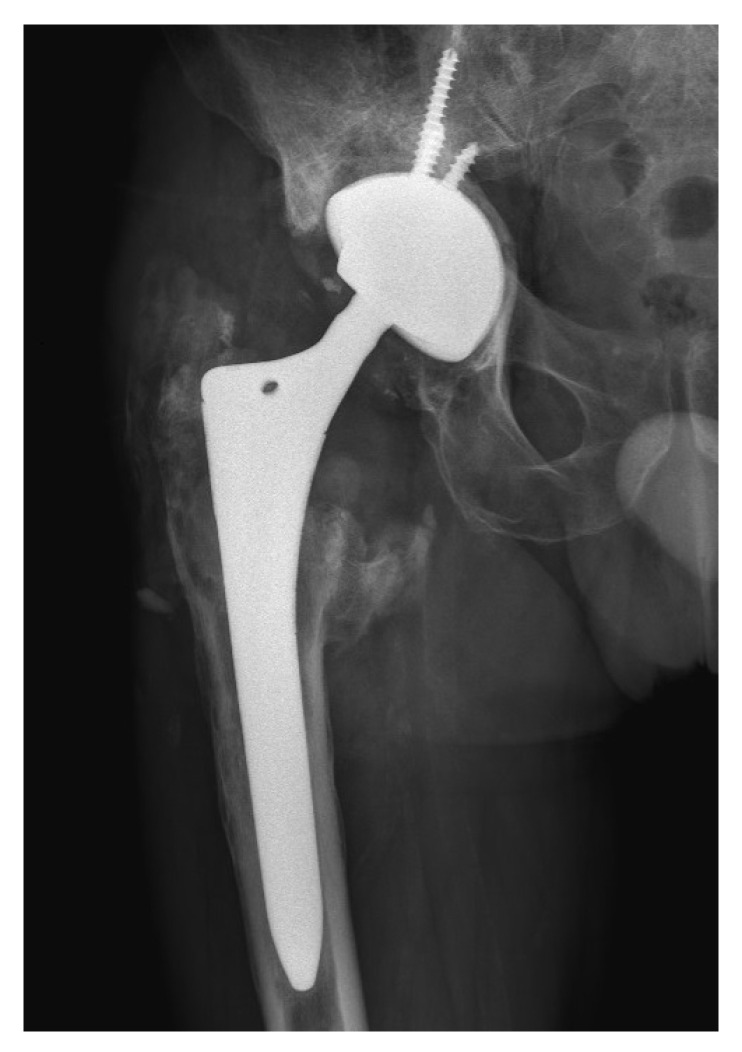
X-ray of the patient’s right hip 6 months after THA revision.

**Figure 6 bioengineering-11-00235-f006:**
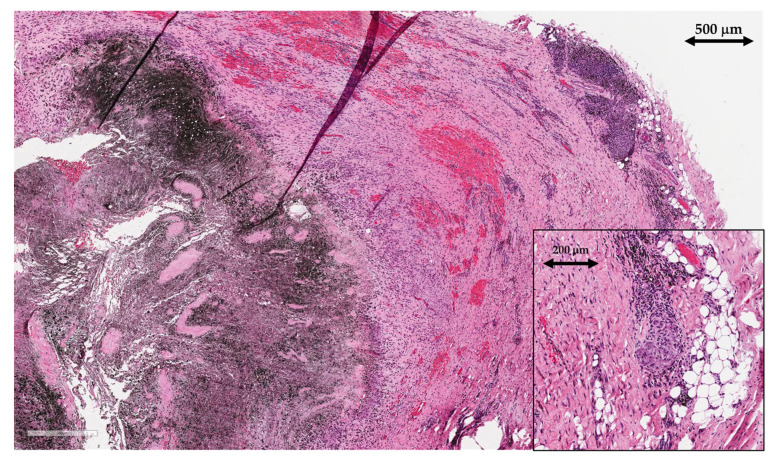
Histopathological findings of synovial-like interface membrane (SLIM), including areas of necrosis with abundant deposits of metallic microparticles and macroparticles (left), surrounded with layers of macrophages. Areas of hemorrhage (middle) and granulomatous inflammation with foreign body giant cells (upper-right corner, magnified in lower-right corner) are also present (scale is written on each part of the image).

**Figure 7 bioengineering-11-00235-f007:**
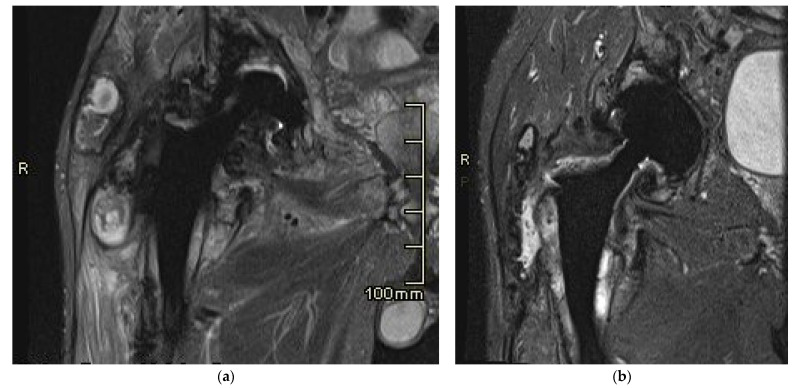
Magnetic resonance imaging (MRI) of the right hip in coronal plane. (**a**) Before revision; (**b**) six months after revision surgery.

**Table 1 bioengineering-11-00235-t001:** Patient’s laboratory findings at revision surgery and at 6-month follow-up.

Investigation	Norm	Unit	At Revision	+6 Months
ESR	<15	mm/h	5	2
CRP	<5	mg/L	4	6
Leukocytes, total	4.00–10.00	10^9^/L	6.93	6.92
Neutrophils, total	1.50–7.40	10^9^/L	NA	4.57
Neutrophils, segmented	<0.03	10^9^/L	NA	0.03
Lymphocytes	1.10–3.50	10^9^/L	NA	1.25
Monocytes	0.21–0.92	10^9^/L	NA	0.75
Eosinophils	0.02–0.67	10^9^/L	NA	0.31
Basophils	0.00–0.13	10^9^/L	NA	0.04
Erythrocytes	4.50–5.50	10^12^/L	5.57	6.21
Hemoglobin	130–170	g/L	154	167
Hematocrit	0.40–0.50	-	0.47	0.51
MCV	83.0–101.0	fL	83.8	82.4
MCH	27.0–32.0	pg	27.6	26.9
MCHC	315–345	g/L	330	326
Platelets	150–410	10^9^/L	545	355
Glucose	3.6–6.1	mmol/L	6.0	5.9
Urea nitrogen	2.8–7.5	mmol/L	2.6	2.1
Creatinine	64–104	μmol/L	46	56
GFR	80–120	mL/min/1.73m^2^	>90	>90
Bilirubin—total	<17	μmol/L	5	12
Bilirubin—direct	<5	μmol/L	2	3
AST	<0.58	μkat/L	0.77	0.54
ALT	<0.74	μkat/L	0.49	0.47
γ-GT	<0.92	μkat/L	0.85	1.05
Sodium	135–145	μmol/L	136	136
Potassium	3.5–5.3	μmol/L	4.91	4.24
Chlorides	97–110	μmol/L	103	106

ESR—erythrocyte sedimentation rate; CRP—C-reactive protein; MCV—mean corpuscular volume; MCH—mean corpuscular hemoglobin; MCHC—mean corpuscular hemoglobin concentration; GFR—glomerular filtration rate; γ-GT—gamma-glutamil-transferaze; AST—aspartate aminotransferase; ALT—alanine aminotransferase; NA—not available.

**Table 2 bioengineering-11-00235-t002:** Patient’s serum trace element concentrations at revision surgery and at follow-up.

Serum Trace Element	At Revision	+4 Months	+6 Months
Titanium *	891.4	1237.6	879.1
Aluminum *	11	NA	15
Vanadium *	6.33	NA	5.38
Cobalt *	<0.07	NA	0.1
Chromium *	<0.257	NA	<0.257

* Values in μg/L. NA—not available.

## Data Availability

The data presented in this study are available on request from the corresponding author (privacy reasons).

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
