# Peer review of "Extreme Serum Titanium Concentration Induced by Acetabular Cup Failure: Unveiling a Unique Scenario of Titanium Alloy Debris Accumulation"

_bioengineering, 2024, doi:10.3390/bioengineering11030235_

Round 1
Reviewer 1 Report
Comments and Suggestions for Authors
Authors reported the extreme serum titanium concentration induced by catastrophic acetabular cup failure: Unveiling a unique scenario of titanium alloy debris accumulation. This paper can be accepted for publication after minor revision.
(1)Please strengthen the literature review with few more references.
(2)Ensure the full form for all the aberrations given in the manuscript.
(3)The objective should be given in details and the novelty of the work is to be mentioned in the introduction section.
(4)The conclusion section does not show impactful findings
(5)Numerical results / findings data is missing in the paper
(6)Additional supporting experiments is required to justify the conclusion.
(7)The discussion related to releasing of titanium alloy debris should be elaborated
(8)The evidence is to be provided for the presence of debris with compositional studies
Reviewer 2 Report
Comments and Suggestions for Authors
Review report
This work is a case study of a total hip replacement. The prosthesis has a ceramic-ceramic articular joint with a titanium acetabular cup, and it fixed by using press-fit method. In this case study, after a few years, the prosthesis suffered wear in the joint and the ceramic head reached the metallic cup and faced it with wear. In this work, revision has been done and the evaluation of the released titanium level has been measured. In my opinion, due to the issues that I will mention in the following, the authors have not been able to present the conclusions based on the obtained results. What the authors have presented as the main approach needs more accurate evaluations.
What is clear is that due to fracture and disintegrate of the ceramic liner, it is possible for the ceramic head to come in contact with the metal cup (and also the presence of ceramic particles in that area as a third-body). Due to the high hardness of alumina ceramic and the low wear resistance of titanium alloy, the metal cup has suffered wear and fragmentation. This fragmentation caused inflammatory reaction as a foreign body reaction and the presence of macrophages and giant cells (FBGCs) in that area. Adverse tissue reactions are due to the presence of a foreign body. Although the fragmentation of metal particles causes an increase in the surface in contact with body fluids and the release of titanium ions, but the increase in titanium concentration is not necessarily the cause of adverse tissue reactions, or at least the results of this work do not show such this statement. This claim could be correct when instead of measuring titanium in the serum, its amount was measured in the surrounding and damaged tissues. Then the authors could attribute the problems that occurred to the accumulation of high concentration of titanium in the tissues.
Overall, in my opinion, the authors should present the results of this case study in such a way that some kind of conclusion has already been drawn.
In addition to the above, below are some statements that need to be refined:
1. In the first and second paragraphs of the introduction, it is not clear to the authors what materials the components of a hip joint prosthesis can be made of, especially in the articular joint part. 2. lines 152-153, What is the relation between the statement mentioned cobalt and chrome and this work? Is related to the result of the same work!Comments on the Quality of English Language
Minor editing of English language required.
Reviewer 3 Report
Comments and Suggestions for Authors
The authors present an interesting case where the ceramic head of a hip prosthesis penetrated through a TiAlV acetabular cup in the absence of the ceramic liner. This led to high Ti-concentrations in the serum and the neighbouring tissue (grey-black), but luckily for the patient, he did not suffer adverse effects besides the failed prosthesis, there was no inflammation.
Title:
Part of the title is misleading and exaggerated. There was nothing wrong with the cup being there, while the ceramic liner wasn't. TAV cups are not meant to be abrasion resistant. Thus, I'd claim others to be responsible for the "catastrophic … failure".
Abstract:
It should be mentioned that there was no liner found.
Line 20: I'd change to "During revision surgery", it was found that the "ceramic head…"
Introduction:
I'm not convinced that the elastic modulus is responsible for the osteoconductivity, since the elastic modulus is much higher than the one of bone.
M&M:
Looking at figure 1 I find (a) that the head is penetrating a lot into the cup and (b) that there is an offset of the ceramic head within the cup. Thus, I really doubt that at this time point there was any liner present in the cup and I'm convinced that the "unusual sounds in the hip during weight-bearing" were from the ceramic head hitting the TAV cup. You know better the geometry of the prosthesis; did you apply a reconstruction of the head position on the X-rays?
Ion analysis:
You hardly provide any information about the ion analysis (except referring to the lab), while in the discussion you write that a comparison of different ion concentrations is difficult. Thus, to be on the forefront, you should provide information about the digestion process and the method being applied to analyse the ion concentration (e.g. ICP-OES, ICP-MS, parameters).
What's the uncertainty of the results you provide? Reproducibility? How many digits after the comma are actually relevant? What's the "norm" you refer to in section 3.1, is this the detection limit? Were Co and Cr below the detection limit?
Discussion:
I'd put the part about the missing liner as the first paragraph (currently the last paragraph). From the X-rays I believe that there has never been a ceramic liner implanted despite the fact that the label was put in the surgery report. It is very unlikely that the "liner was ground into tiny and fine-grain debris" because then the head, which was made of the same material, would have been ground to the same fine-grain debris as the liner. And if it was broken, you would have found residues of it.
Conclusions:
Please add the main points of your Method, Results and Discussion.
General question:
Are there news from the 1-year follow-up in the mean time?
Author Response
Please see the attachmet.

Reviewer 4 Report
Comments and Suggestions for Authors
This article enumerates the effects of failure of the arthroplasty implants.
Line 69, what is 0 mm?
In Fig 2, the abnormal regions may be highlighted for readers to clearly see and avoid searching.
Figure 3 shows a hole created by the alumina ceramic head on the titanium cup? this is unusual. Author should give a detailed literature on what is the statistics of such failure occured. This is questioning a tried and tested procedure of hip replacement material
Line 202... "Patients 201 undergoing hip replacement with loosened titanium components commonly exhibit elevated serum titanium levels at the time of implant failure.... why does this happen?
Line 217, ........" Unlike cobalt, titanium accumulates in organs, posing a risk of tissue injury".... still why titanium is used as implants?
The conclusion is inconclusive.
This article is a mere case study rather than a research article.
Reviewer 5 Report
Comments and Suggestions for Authors
This is a very valuable case of implant accidents and complications for bone implant patients 14 years later. This case is of great significance for guiding the design and use of bone implants, although only one example is shown. Of course, it would be even better if the blood test results could be presented in the article, and the correlation between the event and the blood results could be discussed. Anyway, I strongly recommend the publication of this paper.
Reviewer 6 Report
Comments and Suggestions for Authors
Dear Editor,
Thank you for inviting me to review the manuscript entitled: “Extreme serum titanium concentration induced by catastrophic acetabular cup failure: Unveiling a unique scenario of titanium alloy debris accumulation” submitted for publication in bioengineering.
The authors, in this context, present a case of unusual primary THA failure where a substantial release of Ti-alloy debris was observed. This case provides an instructive illustration for studying the consequences of substantial release of Ti-alloy debris from orthopaedic implants.
In particular:
The abstract is well-written providing a summary of the case and the importance of it.
The introduction provides the basic information about the coming case. Please add a small paragraph about the impact of age and gender in THA revision, loosening and debris. (eg Younger males have higher revision rates).
Rename the section “Materials and Methods” to “Case Presentation”. In figure 1 the authors state that standard radiographs of the right hip were deemed normal, as depicted in the first available postoperative X-ray (Line 73). I would notice some heterotopic ossification in the level of the lesser trochanter. Is that something that was discussed? Please mention this and add a small comment.
Figure 2 is very impressive, depicting the wear of the materials. Good illustration and presentation. In line 93 the authors state “…for an early revision”- after 17 years from the initial THA this can not be by any means be characterized as early revision. Please rephrase.
I would also erase the subheading “results” and keep all the information under “Case presentation”.
Very good depiction of the histology and the worn materials. I was wondering if there are any intraoperative pictures showing the pseudotumor? Of course this may not be available; nevertheless, it would be nice to have.
The discussion is, in general, well-written. I would also make some comments here regarding the possible risk factors for early material wear. I believe this is the information that is missing.
Round 2
Reviewer 2 Report
Comments and Suggestions for Authors
All comments have well addressed in the new version.
Reviewer 3 Report
Comments and Suggestions for Authors
The paper has improved and is fine now.
Reviewer 4 Report
Comments and Suggestions for Authors
Thank you for inviting me to re review this article. The author have tried to explain the rarity of the case that the absence of the ceramic liner can lead to a worn out femoral component. To confirm that the titanium, can get worn out without the ceramic head, you do not need to study a hip implant, a simple pin on disc study would suffice, and I believe there is a lot of literature available to enumerate on this. Therefore this study may be just a study performed for the sake of performing and does not add any significant information to the scientific community.